# Long-term fluorescence live imaging of honeybee embryos using light sheet fluorescence microscopy and halocarbon-based liquids

Oksana Netschitailo[1,2,*], Paul Siefert[2], Markus M. Middeke[1,2], Artemiy Golden[1], Christin Schrod[1,2], Tim Beringer[1,2], Bernd Grünewald[2], Ernst H. K. Stelzer[1] and Frederic Strobl[1,*]

## ABSTRACT

The eusocial honeybee *Apis mellifera* is a key pollinator and model for insect development, offering insights into the evolutionary history of holometabolous insects. Honeybee embryos develop in a controlled hive environment, which has led to potential adaptations in embryogenesis compared to solitary insect species such as the fruit fly *Drosophila melanogaster*. However, previously applied static imaging techniques are not well-suited to study morphogenic events. Here, we combine mRNA-based transient fluorescence labeling, Perfluorodecalin as the imaging medium, and a custom sample chamber for light sheet fluorescence microscopy to enable long-term live imaging of honeybee embryos from blastoderm formation to hatching. Our approach provides the first dynamic visualization of extra-embryonic membrane formation in honeybees, which exhibits variable window closure locations at the posterior-ventral area of the embryo. This contrasts with the red flour beetle *Tribolium castaneum*, where the serosa window closes at a confined anterior-ventral area. Taken together, our methodological framework expands the toolkit for alternative insect models, enabling comparative studies and investigations of environmental stressors, such as pesticides, on development. Plasmids used and datasets acquired in this study are publicly available, supporting future studies on insect diversification and conservation.

KEY WORDS: Insects, Honeybee (*Apis mellifera*), Morphogenesis, Extra-embryonic membranes, Light sheet fluorescence microscopy, Live imaging

## INTRODUCTION

Honeybees have been extensively studied due to their economic value as pollinators for agriculture and their eusocial lifestyle with a broad repertoire of social behaviors, remarkable learning capacities and intricate communication (Klein et al., 2007; Tautz, 2008). From a developmental biology perspective, honeybees are particularly intriguing (Hu et al., 2019a): as members of the Hymenoptera, which diverged from other holometabolous insects around 345 million years ago, honeybees represent the earliest branching lineage of this group (Misof et al., 2014). This renders them a key representative for studying the evolutionary processes that shaped the developmental diversification of this clade. For instance, although the western honeybee *Apis mellifera* is classified as a long germ-band insect like the primary insect model organism, the fruit fly *Drosophila melanogaster*, the respective genetic components for embryonic axis patterning differ in their expression or are missing entirely in one or the other species (Cridge et al., 2017; Dearden et al., 2006).

Unlike offspring of most solitary living insect species, honeybee embryos develop in a comparatively sheltered environment: the hive provides stable humidity and temperature (Human et al., 2006; Kleinhenz et al., 2003; Siefert et al., 2021; Stabentheiner et al., 2021) while worker bees feed and clean the larvae, combat parasites, and protect the nest with propolis, which has antibiotic properties (Cremer et al., 2007; Drescher et al., 2017; Seidel et al., 2008). Whether this protective setting has led to evolutionary adaptations to embryonic development has long been a subject of research, with several studies on honeybee morphogenesis conducted throughout the 20th and early 21st centuries (Cridge et al., 2017; Dearden et al., 2009; DuPraw, 1967; Fleig and Sander, 1985, 1986, 1988; Milne et al., 1988; Nelson, 1915). However, these early investigations have been limited by the technology of their time, and therefore no high-resolution live imaging data are available. Hence, key aspects of embryonic development remain insufficiently understood, such as the formation of extra-embryonic membranes (EEMs), the cell-based epithelia that provide active protection to the embryo (Schmidt-Ott and Kwan, 2016). This stands in strong contrast to, e.g., the red flour beetle *Tribolium castaneum*, where early electron microscopy studies regarding EEM dynamics (Handel et al., 2000) were later complemented by fluorescence live imaging (Benton, 2018; Benton et al., 2013; Jain et al., 2020).

This approach, however, requires the presence of fluorophores within the specimens. A common strategy, successfully applied in several insect models, is the creation of fluorescent protein-expressing transgenic lines (Caroti et al., 2015; Nakamura et al., 2010; Sarrazin et al., 2012; Strobl et al., 2022). Although a germline transformation protocol for honeybees has been established (Schulte et al., 2014; Wagner et al., 2022), this experimental route requires substantial effort and infrastructure as honeybee colonies cannot be maintained in standard biosafety laboratories in the same way as basically all solitary living insect model organisms (Inglis, 2009, 2010). Instead, dedicated flight rooms are necessary to support colony health and reproduction, but these remain technically demanding due to challenges in maintaining stable climate control, simulating natural light and photo periods and

[1]Physical Biology/Physikalische Biologie (IZN, FB 15), Buchmann Institute for Molecular Life Sciences (BMLS), Cluster of Excellence Frankfurt – Macromolecular Complexes (CEF – MC), Goethe-Universität – Frankfurt am Main (Campus Riedberg), Max-von-Laue-Straße 15, D-60438 Frankfurt am Main, Germany. [2]Neurobiology of the Honeybee/Neurobiologie der Honigbiene (IZN, FB 15), Institut für Bienenkunde Oberursel, Goethe-Universität - Frankfurt am Main, Karl-von-Frisch-Weg 2, D-61440 Oberursel, Germany.

*Authors for correspondence (oksana.netschitailo@physikalischebiologie.de; frederic.strobl@physikalischebiologie.de)

O.N., 0000-0002-5059-3022; P.S., 0000-0003-3732-4466; F.S., 0000-0002-5350-0194

ensuring adequate resource supplementation in an enclosed environment (Czoppelt et al., 1980; Pernal and Currie, 2001; Poppy and Williams, 1999).

An alternative is the injection of fluorescent protein-encoding mRNA, a strategy successfully applied across diverse metazoan species, including the starlet sea anemone *Nematostella vectensis* (DuBuc et al., 2014), the tropical marina sea anemone *Aiptasia sp.* (Jones et al., 2018), the amphipod crustacean *Parhyale hawaiensis* (Gerberding et al., 2002), the European lancelet *Branchiostoma lanceolatum* (Hirsinger et al., 2015), the zebrafish *Danio rerio* (Keller, 2013) and the African clawed frog *Xenopus laevis* (Woolner et al., 2010). However, among the insects, nearly all imaging studies in the fruit fly rely on transgenic lines, and to our knowledge the red flour beetle is the only species with a well-established mRNA-based fluorescence labeling protocol (Benton et al., 2013).

In this study, we provide a methodological framework for mRNA injection-based long-term fluorescence live imaging of embryos from the western honeybee *Apis mellifera* using light sheet fluorescence microscopy. Firstly, we compare two injection strategies with focus on embryo survival and fluorescence intensity. Secondly, we identified Perfluorodecalin (PFD), a halocarbon-based liquid with high oxygen-binding capabilities, as a suitable medium for live imaging. Thirdly, we present a novel sample chamber design optimized for honeybee embryos. We applied this framework to record multiple datasets of honeybee embryogenesis starting at the blastoderm stage and lasting up to the moment of hatching, allowing us to compare honeybee EEM formation dynamics with those of the red flour beetle.

## RESULTS
### mRNA injection-based fluorescence labeling of honeybee embryos

As the first step in our mRNA injection-based fluorescence labeling approach, we created the pBSII-Actin°NB-mEmerald(AM) plasmid (Fig. S1A), which can be used for *in-vitro* synthesis of mRNA encoding mEmerald-labeled (Shaner et al., 2005) anti-actin nanobodies (Melak et al., 2017). To introduce the synthesized mRNA into honeybee embryos, we compared two injection strategies: one targeting the anterior-ventral region and the other targeting the mid-ventral region. In general, injection followed established routines for gene manipulation studies (Beye et al., 2002; Otte et al., 2018; Schulte et al., 2014). For our approach, we opted for a mRNA concentration of ~240 pg per embryo, which has been shown to have no adverse effects on embryonic development, even when injected at early stages (Otte et al., 2018), and introduced the mRNA 00:30–03:00 h after egg laying, prior to the onset of cleavage and cellularization (03:30 h and 10:00 h after egg laying, respectively) into the honeybee embryo (DuPraw, 1967; Schnetter, 1934). We observed no significant difference in hatching rates between the injection strategies (63%±12% for the anterior-ventral region compared to 40% ±11% for the mid-ventral region). To determine the optimal approach for long-term live imaging, we imaged injected embryos from the lateral view at four distinct developmental stages using a sample chamber-based light sheet fluorescence microscope (Fig. 1B, Fig. S2) and quantified fluorescence intensity distribution along the anterior-posterior axes. For both strategies, we found relatively strong and

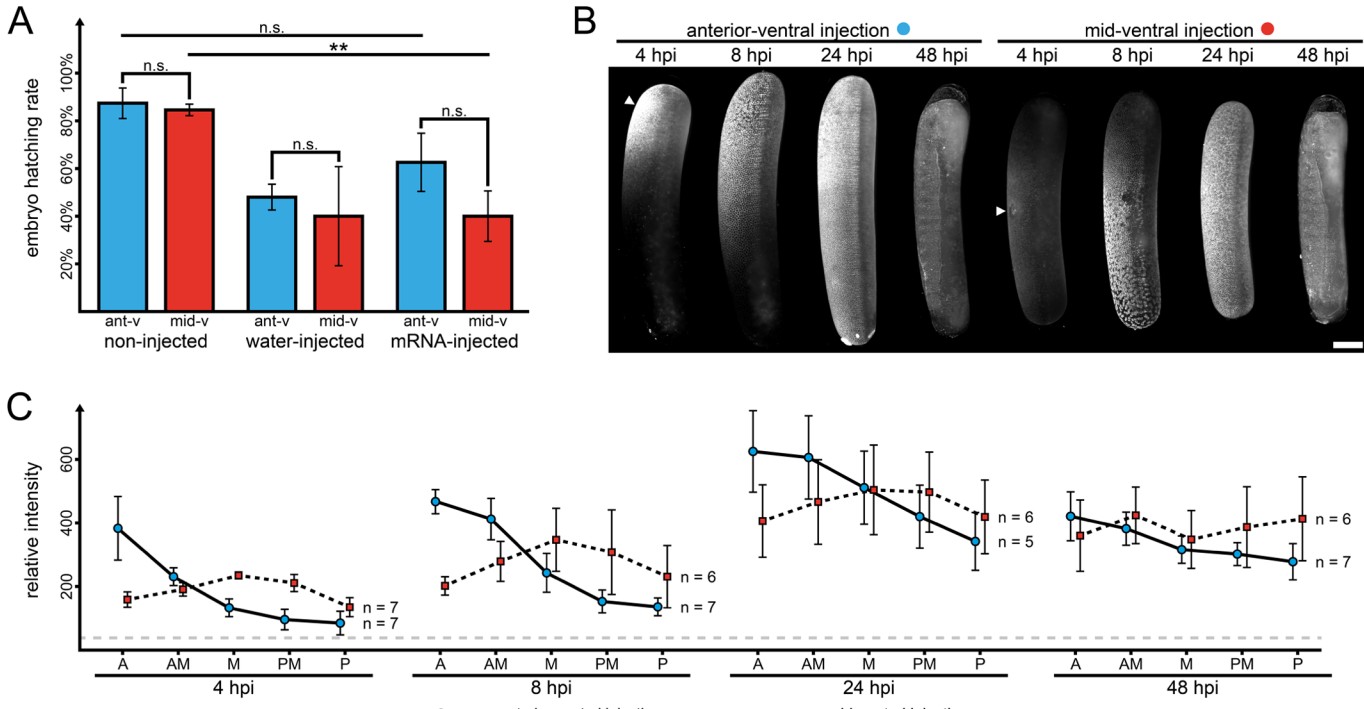

**Fig. 1. Fluorescent labeling of honeybee embryos via mRNA injection**. (A) Embryo hatching rate (*n*=3, 50 embryos injected per condition and repetition) after injection of H₂O or mRNA at either the anterior-ventral (ant-v, blue) or the mid-ventral region (mid-v, red). Two non-injected controls were defined, one from the anterior-ventral and one from the mid-ventral injected embryo batch. In comparison to the controls, the mid-ventral injection approach showed a significantly reduced embryo hatching rate (two-sided Student's *t*-test, **P<0.01). Error bars indicate standard deviations. n.s., not significant. (B) Fluorescence signal distribution at different developmental stages after injection (denoted as hours post injection, hpi) at either the anterior-ventral or the mid-ventral region (white arrowhead). Shown are *z* maximum projections from the lateral view. In both approaches, roughly uniform fluorescence distribution was obtained at 24 hpi and 48 hpi. Scale bar: 200 µm. (C) Quantification of fluorescence intensity at five distinct regions along the anterior-posterior axis (A, anterior; AM, anterior-medial; M, medial; PM, posterior-medial; P, posterior) at different developmental stages. Fluorescence was rather weak and spatially confined at 4 hpi and 8 hpi after injection but became stronger and more homogeneous at 24 hpi and 48 hpi. The gray dashed line indicates background intensity. Error bars indicate standard deviations.

Biology Open

homogenous signal at 24 h post injection (hpi), i.e. at the uniform blastoderm stage, which decreased only marginally until 48 hpi (Fig. 1C). Since no considerable differences in fluorescence intensity and distribution were found at 24 hpi, but hatching for the mid-ventral mRNA-injected embryos was significantly lower compared to the respective non-injected controls (P<0.01, Fig. 1A), we proceeded with the anterior-ventral injection strategy.

## Perfluorodecalin as a suitable imaging medium for honeybee embryos

Previous studies have shown that, under natural conditions, honeybee embryos develop best at 90–95% relative humidity, and that development is impaired in drier and fully saturated atmospheres (Doull, 1976). Imaging in air, however, is disadvantageous due to the large refractive index difference between air (~1) and biological tissue (typically between 1.35 and 1.40) (Gul et al., 2021). Hence, in most fluorescence microscopy assays, the specimens are submersed in aqueous liquids or halocarbon-based oils (Boothe et al., 2017). To identify a suitable imaging medium for honeybee embryos, we compared survival of undisturbed embryos in a wax comb (99% hatching rate) and embryos removed from the comb and incubated in air with ≥80% relative humidity (90% hatching rate) with those incubated in various liquids (Fig. S3). Autoclaved tap water (refractive index 1.333 at 34.5°C), which has been shown to work well for fruit fly (Keller et al., 2011) and red flour beetle embryos (Ratke et al., 2020), led to a strong drop in survival (41% hatching rate). These results are consistent with previous studies indicating that honeybee embryos require gas exchange for proper development (Mackasmiel and Fell, 2000). Previous research also reported that honeybee embryos develop when immersed in halocarbon oil (Milne et al., 1988). However, in Halocarbon Oil 27 (refractive index 1.405 at 34.5°C), hatching was severely impaired (24% hatching rate), consistent with the findings in the cited study. Next, we tested PFD (refractive index 1.310 at 34.5°C), another halocarbon-based liquid, which has been used in plant-associated fluorescence live imaging (Littlejohn et al., 2010) and is known for its high gas solubility (Lowe, 1987). Unlike Halocarbon Oil 27, PFD is fully fluorinated and thus chemically highly inert, and its viscosity is roughly an order of magnitude lower, reducing resistance when moving mounted embryos through the imaging medium in sample chamber-based light sheet fluorescence microscopes. High survival (89% hatching rate) was achieved using PFD, comparable to incubation in air with ≥80% relative humidity.

## Optimized sample chamber design for honeybee embryo live imaging

Using PFD as the imaging medium introduces two new challenges: its low surface tension (17.6 mN/m, i.e. about four times lower than that of water) renders it prone to leakage, and its high volatility poses the risk of specimens 'falling dry' during imaging. Moreover, unlike the fruit fly and the red flour beetle, honeybee embryonic development is confined to a narrow temperature window of ~33–36°C (Harbo and Bolten, 1981; Kleinhenz et al., 2003; Stabentheiner et al., 2021), which is considerably above standard laboratory temperature. To address these issues, we optimized the sample chamber design of our light sheet fluorescence microscope (Fig. S4). Firstly, we reshaped the chamber geometry to reduce the open surface area, thereby minimizing PFD evaporation. Secondly, we introduced custom-made silicone rubber seals at the bottom, top and at the detection objective insertion side of the chamber prevent leakage while still allowing smooth movement and rotation of the sample holder. Finally, temperature control is achieved by a heating foil inserted into a water-filled 'heating cabinet' at the base of the chamber.

## Long-term fluorescence live imaging of honeybee embryos

Based on the outlined technical implementations, we imaged eighteen honeybee embryos. Of these, eight (45%) developed without visible damage, resulting in at least two datasets for each, the ventral and both lateral perspectives (Fig. 2A, Movies 1–3). Larval hatching was confirmed in five embryos (28%), including the embryo from DS0001, where hatching occurred at the end of the imaging process (Fig. 2B, Movie 2). In the image data, we were able to identify specific developmental stages (Fig. 2C) that have previously been defined based on transmission light microscopy (DuPraw, 1967). Examination of optical sections shows that our method provides high-quality data at sub-cellular resolution (cf. Fig. 2A, detail image) in which distinct structures and processes can be identified, such as the convergence of the lateral ectodermal tissue sheets above the mesodermal layer during gastrulation (Fig. 2D, Fig. S5).

## Comparison of honeybee and red flour beetle extra-embryonic membrane formation

Our fluorescence live imaging data provides the first dynamic representation of EEM formation in the honeybee and shows complementary consensus with previous descriptions derived from manual observation (Nelson, 1915) as well as scanning electron microscopy-based still images (Fleig and Sander, 1986). In brief, the EEM differentiates primarily at the dorsal site of the blastoderm; its rims then migrate ventrad over the surface of the emerging germband. The migration process is bidirectional, but asymmetric as the anterior rim migrates at a higher velocity and covers a considerably larger distance than the posterior rim, leading to the formation of an EEM window at the posterior-ventral region of the embryo (Fig. 3A, 13:30–16:00 h), which eventually closes (Fig. 3A, 16:30–17:00 h).

To interpret this morphogenic process in a phylogenetic context, we collected live imaging data from red flour beetle embryos using the transgenic AGOC{Zen1'#O(LA)-mEmerald} #1, #2 and #3 sublines (Strobl et al., 2018), in which filamentous actin is specifically labeled in the serosa, one of the two EEMs in this species. The serosa emerges in the anterior-ventral, the lateral and the posterior-dorsal regions of the blastoderm, and migrates towards the ventral side, leading to the formation of the serosa window at the anterior-ventral region (Fig. 3B).

Juxtaposition of window closure locations suggests considerably greater spatial variation in the honeybee than in the beetle (Fig. 3C, Fig. S6). We quantified closure locations and assessed their relationship to embryo size. In both species, linear regression indicated that larger embryos tend to close the window more medio-ventrally (P<0.05 for both, Fig. 3D). However, variation in the honeybee was about 2.3 times higher than in the red flour beetle (standard deviations of ~7.3% and ~3.1%, respectively).

## Multicolor labeling and multi-channel imaging of honeybee embryos

In addition to the anti-actin nanobody-encoding plasmid, we designed nine open-access plasmids for in-vitro mRNA synthesis that allow labeling of nuclei (via the SV40 nuclear localization sequence, NLS), filamentous actin (via Lifeact, LA), or cell membranes (via the GAP43 membrane anchor, MEME), in blue, green, and red (Table S3). We validated the performance of three of these plasmids in a co-injection assay, demonstrating their suitability for multi-color labeling and multi-channel imaging of honeybee embryos (Fig. 4). To obtain a more comprehensive picture, we imaged the embryo along four directions, revealing uniform fluorescence on the ventral, lateral and

Biology Open

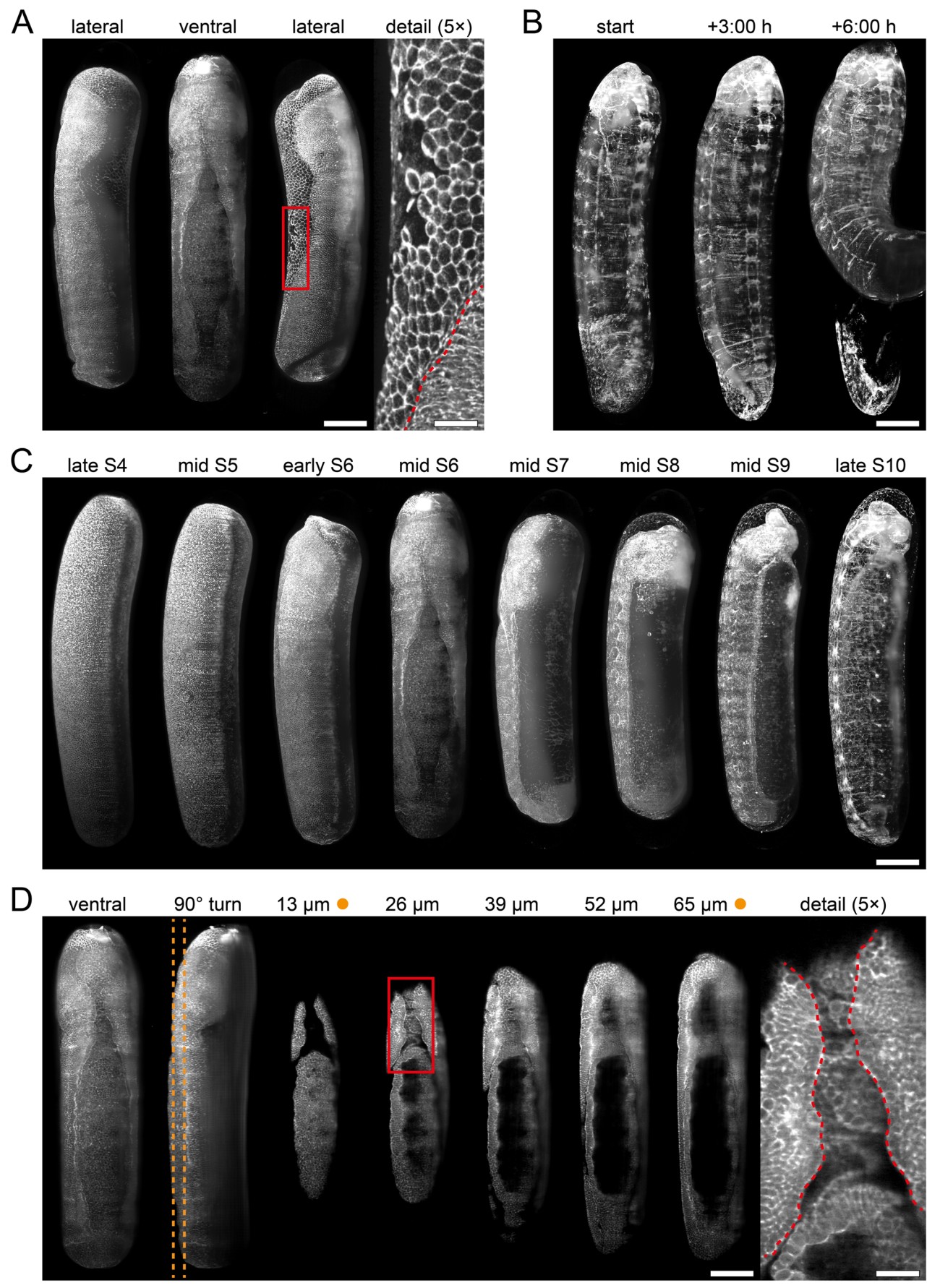

**Fig. 2.** See next page for legend.

**Fig. 2. Fluorescence live imaging of honeybee embryos injected with mRNA encoding mEmerald-labeled anti-actin nanobodies using light sheet fluorescence microscopy.** (A) Three embryos shown from different views during gastrulation. The detail image (position outlined by the red rectangle) highlights the boundary (red dashed line) between the extra-embryonic (upper left region) and embryonic tissues (lower right region). (B) Embryo hatching in the sample chamber at the end of the developmental period. (C) Embryos at specific developmental stages defined by DuPraw (1967). (D) Optical sections of an embryo along the ventral-dorsal axis during gastrulation. Distances indicate depth from the embryonic surface. Orange dashed lines indicate the locations of the optical sections shown in the third and seventh column (marked by orange dots). The detail image (position outlined by the red rectangle) shows the complex folding of the mesodermal tissue area between two epidermal sheets (red dashed lines). Scale bars: 200 µm (main images) and 40 µm (detail images).

dorsal sides (Fig. S7A). As expected (Engelbrecht and Stelzer, 2006), orthogonal projections along the $x$ axis showed reduced axial resolution, but signal intensity was still sufficient to resolve, e.g., the nuclei (Fig. S7B).

## DISCUSSION
In this study, we provide the first protocol for fluorescence live imaging of honeybee embryogenesis based on three columns: (i) a robust mRNA injection strategy with acceptable survival rate leading to homogeneously strong signal, (ii) the identification of PFD as a suitable imaging medium for honeybee and possibly insect embryos from other species that also do not properly develop in aqueous media, and (iii) a novel sample chamber design that facilitates imaging. This advance enables dynamic 3D investigations of honeybee morphogenesis, paving the way for quantitative and comparative analyses in a key insect model. For convenience, we provide the image data (Table S1) as well as the non-nanobody-based open-access *in-vitro* mRNA synthesis plasmids (cf. Table S2) as resources for the community. In the following sections, we discuss the limitations of our approach, the potential biological context of our findings on EEM formation, and application perspectives of our methodological framework.

### Limitations of our framework
As summarized in Table S3, low embryo survival remains a challenge, despite the various optimization steps implemented in this study. This is most evident across the complete workflow, where multiple factors (injection, mounting and light sheet fluorescence microscopy-based imaging in PFD) contribute cumulatively: only about half of the embryos develop for multiple hours without visible damage, and just a quarter hatch during imaging. Both injection as well as incubation in PFD reduce survival, and it is conceivable that the embryos are further stressed by the laser light irradiation, a factor that we did not independently assess due to the high technical effort. In consequence, although we set up multiple recordings to image embryos from the dorsal view, we were not able to obtain a dataset in which one of these embryos survived until the end of the imaging period [for reference, we provide one dataset of an embryo from the dorsal view that shows signs of tissue disintegration (DS0004)]. Even though we followed a more conservative long-term live imaging strategy for the honeybee embryos by imaging them only along one direction, the hatching rate stands in strong contrast to the assayed red flour beetle embryos, where all nine specimens hatched, even though they were imaged along four directions. The evidenced low tolerance of honeybee embryos to stressors can probably be explained by their eusocial lifestyle. Honeybee colonies keep eggs in highly protected, environmentally controlled areas within the hive

(Human et al., 2006; Kleinhenz et al., 2003; Siefert et al., 2021; Stabentheiner et al., 2021). In consequence, honeybee eggs have likely experienced lower selection pressure to withstand adverse environmental conditions compared to solitary living insect species such as the red flour beetle (Donoughe, 2022).

A key limitation of fluorescent labeling via mRNA injection is its restriction to ubiquitous labeling. By contrast, tissue-specific expression can only be achieved through transgenesis, e.g. by designing expression cassettes that utilize tissue-specific promoters. However, the difficulties of this experimental route are outlined in the Introduction section. Additionally, our study is limited to morphogenic information from the female sex, as we refrained from imaging of male honeybee embryos due to their lower survival rates during injection (Gempe et al., 2009). While fertilized eggs develop into workers or queens, unfertilized eggs, which are longer and exhibit slightly slower development, give rise to drones (Wegener et al., 2010). Hence, further comparative investigations are needed to determine how these morphological and temporal differences are reflected during embryonic development.

Lastly, the presented approach is limited in throughput. With the current cobweb holder design, only a single honeybee embryo can be mounted and imaged at a time, whereas the smaller size of red flour beetle embryos allows simultaneous imaging of up to three specimens (Ratke et al., 2020). Throughput can be increased by modifying the cobweb holder to include multiple slotted holes aligned along the $y$ axis. However, the number of embryos that can be imaged simultaneously in sample chamber-based light sheet fluorescence microscopes would still be constrained by other factors such as the maximum travel range of the microtranslation stage along $y$. Achieving high-throughput imaging may therefore require alternative solutions such as open-top light sheet setups (Moos et al., 2024), although these devices come with their own limitations.

### Honeybee extra-embryonic membrane formation in an evolutionary context
In this study, we demonstrated that in both the honeybee and the red flour beetle, larger embryos tend to close their EEM windows more medio-ventrally. However, variability in closure location was more than twice as high in honeybee embryos, with one case even occurring 'beyond' the posterior pole (Fig. S6B, first column). This difference may relate to a specific feature of EEM formation in the beetle: anchoring of the serosa to the vitelline membrane during gastrulation (Münster et al., 2019), which likely restricts closure to a narrow target area. Speculatively, differences in closure location variability could also reflect differences in EEM degradation, which follows a complex cascade in the beetle (Panfilio et al., 2013) but appears simpler in the honeybee (DuPraw, 1967; Nelson, 1915). This raises the possibility that a simpler degradation process requires less spatial precision during setup, although additional experiments are required to test this theory.

### Perspective
For the honeybee, as a key representative of the Hymenoptera, our methodological framework serves as gateway for advanced morphogenesis-related research. CRISPR/Cas9 gene editing has been established in the honeybee (Kohno and Kubo, 2018; Kohno et al., 2016) and optimized over the last years to enable gene function studies (Hu et al., 2019b). Combination of both approaches by double injection of both the fluorescent protein-encoding mRNA in conjunction with the gene editing mixture, consisting of sgRNA and Cas9 protein (Hu et al., 2019b; Nie et al., 2021) will allow to

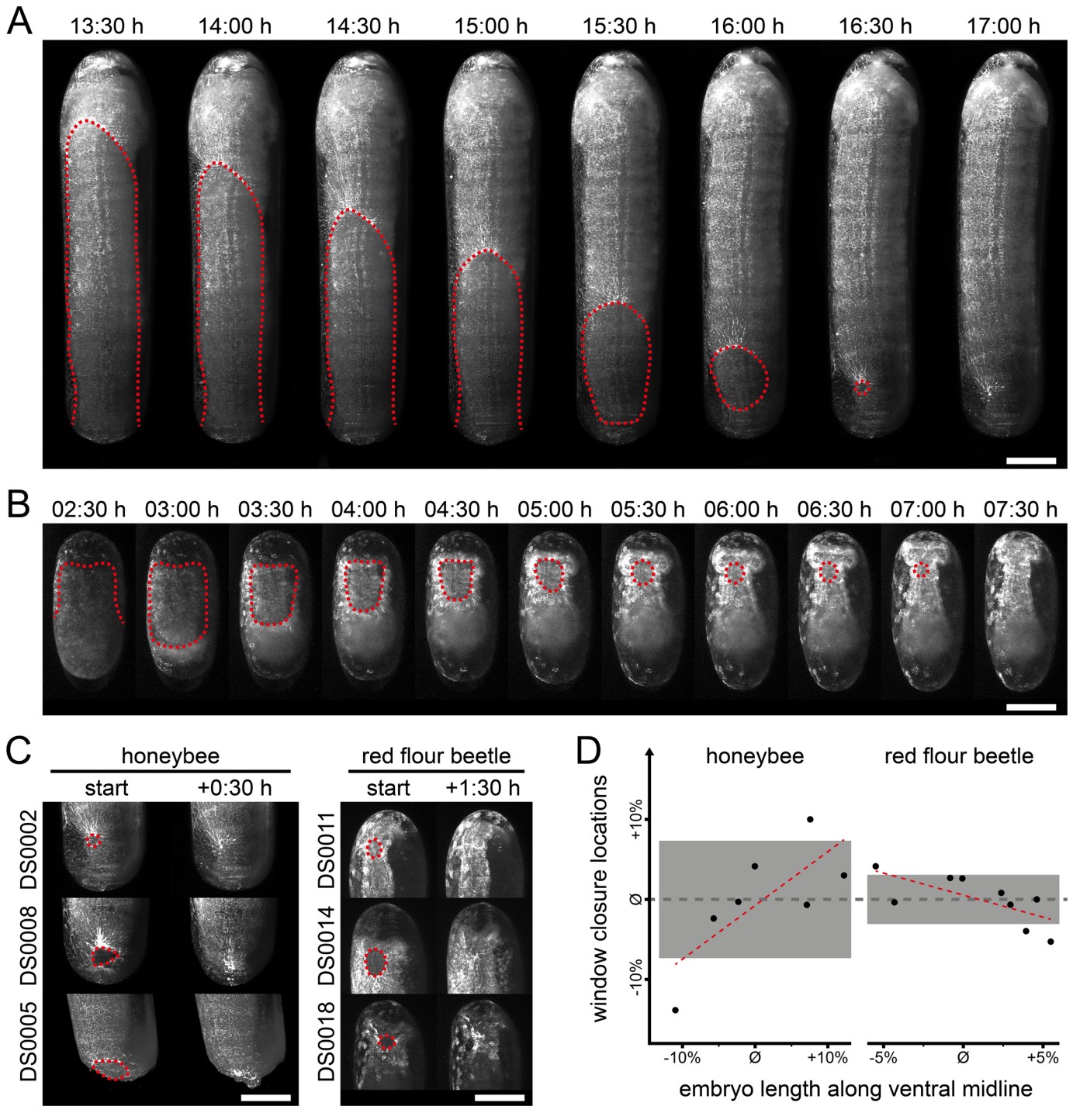

**Fig. 3. Comparative EEM formation dynamics during embryogenesis.** (A) EEM window closure during honeybee embryogenesis. The EEM emerges in the anterior-ventral region of the embryo and extends towards the posterior pole (13:30–15:00 h). After crossing the median, the EEM proceeds to close in at the lateral and dorsal sides, giving rise to a window in the posterior-ventral region of the embryo (15:30 h). The window becomes smaller (16:00–16:30 h) until eventual closure (17:00 h). (B) Serosa window closure during red flour beetle embryogenesis. The serosa emerges in the anterior-ventral, the lateral and the dorsal regions of the embryo and extends from there (02:30 h). After migration of the posterior pole (03:00 h), serosa window emerges and closes in the anterior-ventral region of the embryo (04:00–07:30 h). (C) While honeybee embryos show variable EEM closure locations, the serosa closes at roughly similar locations in red flour beetle embryos. Scale bars: 200 μm. (D) Window closure locations along the ventral midline from the posterior pole (0%) to the anterior pole (100%) plotted relative to the average closure location (i.e. 5.5%±7.3% for the honeybee, 69.6%±3.1% for the red flour beetle; gray dashed line) as a function of embryo length relative to the average embryo length. Light gray rectangles indicate standard deviations; red dashed lines are best-fit lines.

observe the effect of altered gene functions dynamically during embryonic development. Since zygote formation is not completed until 1:30–2:00 h after egg laying (Yu and Omholt, 1999), full knockouts can be achieved in the honeybee if injection is performed before that developmental stage is reached. The anterior-ventral injection strategy, which we favor due to the higher survival rate, is also advantageous for such experiments, as it targets the region in which the pronuclei fuse to give rise to the zygote.

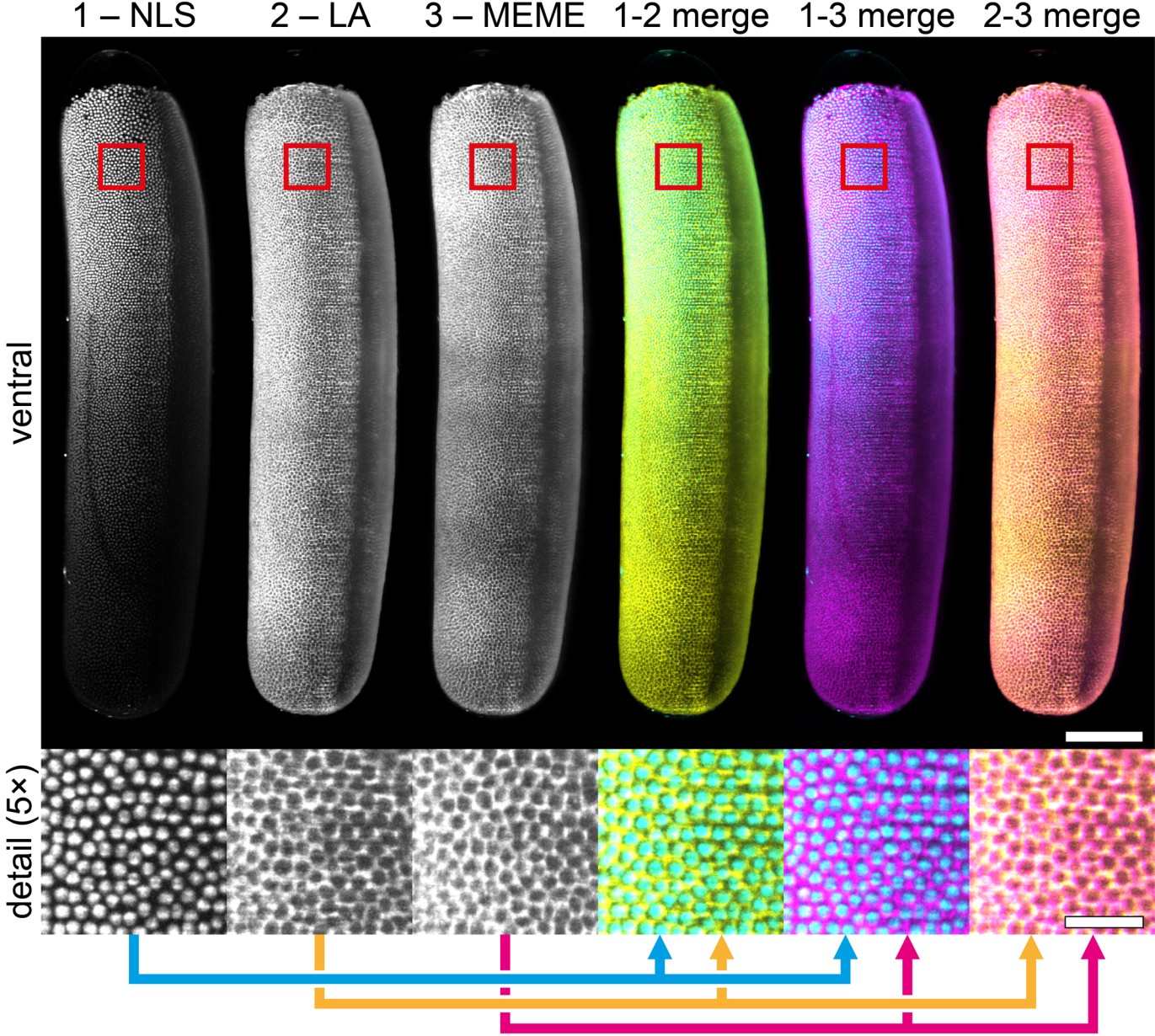

**Fig. 4. Multicolor imaging of honeybee embryos.** Embryo at the uniform blastoderm stage injected with a mRNA mixture encoding the mTagBFP2-labeled NLS tag, the mEmerald-labeled LA tag, and the mCherry-labeled MEME tag. Filamentous actin, marked by mEmerald-labeled LA, is mainly found at the lateral cell borders, as expected in this developmental stage. Red rectangles outline the positions of the detail images. Scale bars: 200 µm (main images) and 40 µm (detail images).

Long-term fluorescence live imaging of honeybee embryos can also be a valuable technique in other disciplines such as ecology and ecotoxicology, e.g. as a novel approach to characterize the effects of chemicals, especially pesticides, on honeybee embryo health. For example, neonicotinoids – neuroactive substances that interact with the acetylcholine pathway (Grünewald and Siefert, 2019) – severely affect orientation, motor function, learning and brood development (Fischer et al., 2014; Siefert et al., 2020; Williamson et al., 2014; Wright et al., 2015), but these substances have never been assayed for their effects on embryonic development due to a lack of suitable methodology. Our approach allows to study even sublethal effects of chemicals on the embryogenesis of honeybees in a dynamic fashion and may thus provide links between abnormal physiology and non-conformal behavior (Alkassab and Kirchner, 2017).

## MATERIALS AND METHODS
### Insect rearing and transgenic lines
Honeybee embryos were produced by naturally mated *Apis mellifera carnica* queens. Colonies were kept in the facilities of the Institut für Bienenkunde Oberursel, Germany. Red flour beetle embryos were produced using the transgenic AGOC{Zen1′#O(LA)-mEmerald} #1, #2 and #3 sublines (Strobl et al., 2018), which express mEmerald-labeled (Shaner et al., 2005) Lifeact (Riedl et al., 2008) under control of the serosa-specific *Zerknüllt 1* (Gene ID: 641533) promoter. Cultures were kept in numbers of 200–500 individuals in growth medium [full grain wheat flour (SP061036, Demeter) supplemented with 5% (wt/wt) inactive dry yeast (62-106, Flystuff)] in one-liter glass bottles in a 12:00 h light/12:00 h darkness cycle at 32°C and 70% relative humidity (DR-36VL, Percival Scientific). All animal-related experiments were approved by the institutional ethics committee (Tierschutzkommission der Goethe-Universität, which is

supervised by the Regierungspräsidium Gießen, Hessen, Germany) and performed in agreement with the German Animal Welfare Act (Tierschutzgesetz/Tierschutz-Versuchstierordnung, based on ETS No.123 (European Convention for the Protection of Vertebrate Animals used for Experimental and other Scientific Purposes) and EU Directive 2010/63/EU (European Directive, 2010/63/EU, 2010)) as well as the ARRIVE 2.0 guidelines (du Sert et al., 2020). Genes were designated according to the Tcas5.2 assembly (Herndon et al., 2020) of the *Tribolium castaneum* genome (Richards et al., 2008).

### Creation of pBluescript II-based plasmids for mRNA synthesis
During this project, ten plasmids suitable for *in-vitro* mRNA synthesis were constructed based on the pBluescript II SK (+) backbone (Stratagene). Among these, the pBSII-Actin°NB-mEmerald(AM) plasmid (cf. Fig. S1A) allows *in-vitro* synthesis of mRNA encoding mEmerald-labeled anti-actin nanobodies (Melak et al., 2017). The other nine open-access plasmids enable *in-vitro* synthesis of mRNA encoding three fluorescent labels [mTagBFP2 (Subach et al., 2011), mEmerald (Shaner et al., 2005), or mCherry (Shaner et al., 2004)] combined with three specific tags [the SV40 nuclear localization sequence (NLS) tag (Sarrazin et al., 2012), the LA tag (Riedl et al., 2008), or the GAP43 membrane anchor with an extended linker (MEME) tag (Benton et al., 2013)} for targeted intracellular localization in all possible combinations (Table S2). A representative plasmid map of the pBSII-MEME-mCherry(AM) plasmid is shown in Fig. S1B. The plasmids were designed for efficiency and ease of use: (i) the coding sequences for the fluorescent proteins were codon-optimized for expression in the honeybee [indicated by '(AM)' in the plasmid names], (ii) the fluorescent proteins were selected to match standard laser lines used in confocal and light sheet fluorescence microscopes (i.e. 405 nm for mTagBFP2, 488 nm for mEmerald, and 561 nm for mCherry), ensuring efficient excitation with minimal spectral crosstalk during multi-channel imaging, and (iii) all plasmids contain the same prokaryotic resistance cassette (ampicillin) and use the same promoter (T7) for *in-vitro* mRNA synthesis, simplifying laboratory routines.

### *In-vitro* mRNA synthesis using the pBSII-based plasmids
*In-vitro* mRNA synthesis was performed with a suitable kit [HiScribe T7 ARCA mRNA Kit (with Tailing), E2060S, New England Biolabs] by following the manufacturer's instructions. Incubation time for the main synthesis was set to 1:00 h. mRNA was purified using a column-based clean-up kit [Monarch Spin RNA Cleanup Kit (50 μg), T2040S, New England Biolabs] according to the manufacturer's instructions with the final incubation time prolonged to 10 min to maximize yield. mRNA was eluted in 20 μl nuclease-free water in average concentrations of 1700 ng/μl and stored at −80°C. For injection, each mRNA was diluted to a concentration of 600 ng/μl.

### Injection of honeybee embryos
Honeybee embryos were collected using dedicated egg collection boxes (Karl Jenter Queen Rearing Kit, Karl Jenter GmbH). Queens were caged in small boxes with plastic combs for 01:30 h. Eggs were collected from the boxes on removable plastic comb plugs at the back of the queen cage. Plugs containing embryos were placed on Petri dishes lines with clay to facilitate orientation of embryos during injection (Fig. S8A–C). Embryos were injected according previously described protocols (Schulte et al., 2014). In short, honeybee embryos were injected 00:30–03:00 h after egg laying with 600 ng/μl mRNA in nuclease-free water for 100 ms with a pressure of 720 hPa using ICSI-micropipettes (with spike, straight, ID: 5 μm, BA: 0°, BL: 0 mm, TL: ~8 mm, PL: 55 mm, glass: BM100T-10P, VICsp-5-0-0-55, BioMedical Instruments), a micromanipulator (MK1, Singer Instruments), and a microinjector (FemtoJet, Eppendorf SE) supplied with pressure through a nitrogen cylinder. Balancing pressure was 50 hPa. The injection parameters resulted in an injection volume of ~400 pl per embryo, corresponding to ~240 pg mRNA. After injection embryos were incubated at 34.5°C in plastic boxes with 0.25 ml 16% sulfuric acid solution per liter air volume (Fig. S8D).

### Light sheet fluorescence microscopy
Light sheet fluorescence microscopy was implemented using a single-sided illumination/single-sided detection digital scanned laser light sheet fluorescence microscope (DSLM) (Keller et al., 2008), which generates a dynamic light sheet by rapidly scanning a Gaussian laser beam with a two-axes piezo-driven scanning mirror (M-116.DG, Physik Instrumente GmbH & Co KG). The customized sample chamber is described in the Results section and depicted in Fig. S4. Illumination was performed through a 2.5× NA 0.06 EC Epiplan-Neofluar objective (422320-9900-000, Carl Zeiss AG) and signal was collected through a 10× NA 0.3 W N-Achroplan objective (420947-9900-000, Carl Zeiss AG) using a high-resolution CCD camera (Clara, Andor). Conventionally (Stelzer et al., 2021), the illumination axis was defined as *x*, the rotation axis as *y*, and the detection axis as *z*. Three micro-translation stages (M-111.2DG, Physik Instrumente GmbH & Co KG) and a precision rotation stage (M-116.DG, Physik Instrumente GmbH & Co KG) were used for sample translation along the *x*, *y* and *z* axes and rotation around the *y* axis, respectively.

### Honeybee and red flour beetle embryo mounting and microscopy
To mount honeybee and red flour beetle embryos, the cobweb holder method was used (Strobl et al., 2017). Injected honeybee embryos were carefully transferred from their respective plastic comb plugs to the thin agarose film covering the slotted hole (2 mm along the *x* axis and 5 mm along the *y* axis) of the cobweb holder using a small paint brush (Fig. S8E). The anterior-posterior axes of the embryos were aligned with the *y* axis of the slotted hole. Depending on their size, honeybee embryos were imaged in either two or three spatially overlapping *z* stacks (also known as 'tiles' and indicated as 'TL' in the file names) along one direction for up to 56:00 h with an interval of 0:30 h. Red flour beetle embryos were collected, prepared, and mounted as described previously (Ratke et al., 2020) and imaged along four directions for up to 50:00 h with an interval of 0:30 h, whereas three embryos, one from each above-mentioned subline, were imaged simultaneously. Comprehensive imaging metadata is provided in Table S4.

### Image data processing
Image processing was performed with Fiji (Schindelin et al., 2012), an ImageJ derivate (Schneider et al., 2012). From all *z* stacks, *z* maximum intensity projections were calculated. For the honeybee-derived data, selected *z* stacks (Fig. S9A) as well as *z* projections (Fig. S9B) were stitched using dedicated software (Preibisch et al., 2009). The resulting images were cropped and intensity-adjusted. For the beetle-derived data, *z* stacks and *z* projections were rotated around the *z* axis to align the anterior-posterior axis with the *y* axis of the image. Rotated stacks and projections were cropped, and projections were intensity-adjusted. Fluorescence intensity was quantified in non-adjusted *z* maximum intensity projections by defining five distinct circular regions ($\sim 5 \times 10^4$ μm² each) evenly spaced along the anterior-posterior axis and calculating the average pixel brightness value within each circle.

### Acknowledgements
The authors thank Sven Plath, Sigrun Becker, Diana Weckeiser, and Sebastian Müller for technical assistance. Access to the light sheet fluorescence microscope was generously provided by the Frankfurt Center for Advanced Light Microscopy (FCAM).

### Competing interests
The authors declare no competing or financial interests.

### Author contributions
Conceptualization: O.N., P.S., A.G., B.G., E.H.K.S., F.S.; Data curation: O.N., T.B., F.S.; Formal analysis: O.N., C.S., F.S.; Funding acquisition: P.S., B.G., E.H.K.S., F.S.; Investigation: O.N., P.S., M.M.M., C.S., T.B., F.S.; Methodology: O.N., P.S., A.G., E.H.K.S., F.S.; Project administration: O.N., P.S., F.S.; Resources: O.N., B.G., E.H.K.S., F.S.; Supervision: O.N., P.S., B.G., E.H.K.S., F.S.; Validation: O.N., P.S., F.S.; Visualization: T.B., F.S.; Writing – original draft: O.N., P.S., F.S.; Writing – review & editing: O.N., P.S., M.M.M., F.S.

### Funding
P.S., B.G., E.H.K.S. and F.S. received funding from the 'RobustNature' Cluster of Excellence Application Initiative provided by the Goethe-Universität. A.G. was funded by the Deutsche Forschungsgemeinschaft (DFG, Graduiertenkolleg iMOL,

GRK 2566, speaker Prof. Dr Achilleas Frangakis) through a doctoral research project awarded to E.H.K.S. E.H.K.S. and F.S. received funding from the Quantitative Structural Cell Biology Projects program (Innovations- und Strukturentwicklungsinitiative 'Spitze aus der Breite'). F.S. received funding from the Add-on Fellowship 2019 of the Joachim Herz Stiftung. Open Access funding provided by by Goethe-Universität – Frankfurt am Main. Deposited PMC for immediate release.

**Data and resource availability**
All relevant data and details of resources can be found within the article and its supplementary information.

**Peer review history**
The peer review history is available online at https://journals.biologists.com/bio/lookup/doi/10.1242/bio.062151.reviewer-comments.pdf

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
