## [Peer Review File · Biology Open]

Long-term fluorescence live imaging of honeybee embryos using light sheet fluorescence microscopy and halocarbon-based liquids

Oksana Netschitailo, Paul Siefert, Markus M. Middeke, Artemiy Golden, Christin Schrod, Tim Beringer, Bernd Grünewald, Ernst H.K. Stelzer and Frederic Strobl
DOI: 10.1242/bio.062151

Editor: Tristan Rodriguez

Review timeline

Original submission:	28 February 2025
Editorial decision:	3 June 2025
First revision:	22 August 2025
Accepted:	26 August 2025

Original submission

First decision letter

MS Title: Long-term fluorescence live imaging of honeybee embryos using light sheet fluorescence microscopy and halocarbon-based liquids

Authors: Oksana Netschitailo, Paul Siefert, Markus M. Middeke, Artemiy Golden, Christin Schrod, Tim Beringer, Bernd Grünewald, Ernst H.K. Stelzer and Frederic Strobl

I have now received all the referees' reports on the above manuscript, and have reached a decision. I am sorry to say that the outcome is not a positive one. The referees' comments are appended below, or you can access them online: please go to.

As you will see, the referees raise some significant concerns about your paper, and are not strongly in favour of publication. Having looked at the manuscript myself, I agree with their views, and I must therefore, reject your paper.

First, please accept my apologies for the long delay in returning this decision to you. I appreciate that you chose to submit your work to us, but have been very delayed in handling your and other manuscripts because of a combination of the challenge of finding enough reviewers (a common problem faced by many journals these days), and because the escalating pressure of the US government against Harvard University since January of this year have disrupted my work on all fronts. including my editorial duties. I am sorry that these things have meant that you have had to wait a longer than acceptable time to hear from me, and would ask you not to view this as any failing on the part of the journal.

Second, in terms of your manuscript, although as you will see the reviewers differ in their views of the suitability for publication of your MS, in my judgement the main reason for this difference is the very valid concern of Reviewer 1. I have also examined the deposited data and agree with Reviewer 1 that the embryos used for analysis do not appear to be developing properly. This is a major flaw that does not have a solution other than generating entirely new datasets, which is beyond the scope of a major revision. If you were to be able to develop new data by imaging new embryos and showing that they were able to complete embryogenesis normally despite the imaging,

then an analysis like the one you have presented here could be appropriate for consideration as a techniques and resources paper, but it would need to be reviewed as a new manuscript.

I do realise this is disappointing news, but the journal receives many more papers than we can publish, and we can only accept manuscripts that receive strong support from referees.

I do hope you find the comments of the referees helpful, and that this decision will not dissuade you from considering us for publication of your future work.

Comments from the Reviewers:

Reviewer 1: To the Editor,

The manuscript submitted here by Netschitailo et al reports the first fluorescent live imaging of honey bee embryogenesis. The movies of Apis embryogenesis generated by the authors are beautiful, but there are significant problems with the work.

I am unable to recommend acceptance of this article due to three major reasons, described below in order of significance.

I initially reviewed this manuscript and came to the two conclusions further below, but before submitting my recommendation, I decided to download some of the authors' data from zenodo to see if I could recommend publishing it as a dataset. The very first dataset I looked at (DS0004 DOI 10.5281/zenodo.13946275) showed that the authors claim that their method "does not impair development" is simply not true. This dataset is the only dorsal view shared by the authors and it shows essentially all of the dorsal and anterior tissue disintegrating. This phenotype is so obvious, it is concerning that the authors chose to ignore it. Furthermore, this is the only dorsal view on zenodo, and also the only view not provided as a supplementary movie with the manuscript. I attach a movie of the timelapse with this review. I did not invest the time to check every timelapse on zenodo, but a closer look at Supp Vid 2 also shows defects at the anterior pole of the egg.

Tissue defects during live imaging happen, especially when establishing a new system, but trust is essential in academia. One must be able to trust that when scientists make statements, and present data to support those statements, they are being truthful. Unfortunately, that is clearly not the case here.

Feedback on the manuscript itself, if one only looks at the results in the manuscript:

First, while this article is submitted as a "Research Report", it is not clear what, if any, research question is answered. The authors state that live imaging is superior to previous fixed light and electron microscopy methods for examining development and morphogenesis, yet they do not demonstrate that benefit in this work. Rather, work from 50 to over 100 years ago (Nelson, 1915; Fleig and Sander 1986; Fleig and Sander, 1988; Du Praw 1967) give far more comprehensive and informative reports of Apis embryogenesis, entirely using fixed methods. Netschitailo et al themselves point this out.

Second, if one considers the manuscript for technical advancements, it also falls short. Most of the results section consists of the authors' descriptions of their use of previously established techniques, yet generally without citing previous work:

- embryonic microinjection as reported in Apis for many years - not cited here
- transient expression for live imaging as reported in many species, e.g. fish, cnidarians, spiders, and beetles, - not cited here
- halocarbon oil for imaging as used for many species, with this particular example also used for plants,
- a lightsheet microscope that the authors mention no details of, but is presumably the same they have used in multiple previous reports.

The genuinely new advances are:

- a custom mounting chamber described in some detail, although this is the norm for the field of lightsheet microscopy
- new plasmids for transient expression and live imaging, although the authors do not describe how these tools improve on those previously created for use in other insects

Reviewer 2: SUMMARY OF THE ADVANCE MADE IN THIS PAPER AND ITS POTENTIAL SIGNIFICANCE TO THE FIELD

This is an excellent manuscript describing a real advance in honeybee embryogenesis; the production of a live imaging system. Honeybees embryos are poorly studied for a number of reasons, 1) the difficulty of preparing them, and 2) the lack of a reasonable live imaging system. The manuscript presented here solves one of those key problems and provides clear and detailed instructions on how to repeat the experiments. This is a significant step forward of studies of honeybee embryos and will, hopefully, provide the impetus for others to begin working with this fascinating and important system.

SUGGESTIONS TO AUTHORS

The manuscript is very well written and contains most of the details I would like to see in such a publication. There are a couple of things I would be keen to see added to help practitioners in this space take up this technology. 1) A table of survival rates of their various treatments, in the supplemental materials. While there are details of survival rate for the various solutions used, it would be useful to bring all of that data together. The expected survival rates are hugely useful for anyone repeating these experiments. 2) A diagram of the humid chambers used to incubate injected embryos would be useful- there is quite a lot of disagreement as to what is the best solution for honeybee embryos. The authors obviously have found a solution, and it would be good to share that.

Reviewer 3: SUMMARY OF THE ADVANCE MADE IN THIS PAPER AND ITS POTENTIAL SIGNIFICANCE TO THE FIELD

In this article, the authors describe the combination of a transient fluorescence labelling method with a suitable mounting medium and a customized, temperature-controlled, leak-proof chamber for time-lapse, light-sheet fluorescence microscopy of live developing embryos from the honeybee *Apis mellifera*. This is an important technical advance that will enable to complement genetic approaches with developmental cell biology studies in this species of high scientific interest.

SUGGESTIONS TO AUTHORS

My major comment is that this study is best suited for the Techniques and Resources section rather than the Research Articles/Reports section of Development. Most of the paper describes the technical aspects (embryo microinjection protocol, mounting medium and customization of the imaging chamber) that had to be optimized for live imaging of honeybee embryogenesis. The cell and tissue dynamics accompanying honeybee embryogenesis are not extracted from these image datasets, except for a brief comparison of the precision in the closure sites of the extraembryonic membranes between honeybee and beetle embryos (see also point 5 below)).

Other comments:

1) In the section "mRNA-injection-based fluorescence labelling of honeybee embryos", please indicate the time of injection and the time of blastoderm cellularization (in hours after egg lay) as these are crucial parameter for homogeneous embryo labelling in transient labelling protocols.

2) In the same section or in the M&M section "Injection of honeybee embryos" explain why you chose a single mRNA concentration of 600 ng/ μ l for injections and whether you tried other

concentrations for stronger and/or more homogeneous labelling. In case of multilabel experiments such as the one shown in Fig. 4, please specify in M&M whether each mRNA was injected at 600 ng/ μ l or at different concentrations. Please give also an estimate of the injected volume per embryo with the needles and microinjection set up used.

3) In the section "Long-term fluorescence live imaging of honeybee embryos", specify that live imaging was performed with single-sided illumination, single-sided detection and with a temporal resolution of 30 min. This information is currently scattered in different parts of the manuscript. Importantly, in the M&M section "Honeybee and beetle embryo mounting and microscopy", give all necessary details about the imaging parameters, such as illumination and detection objectives used, light-sheet thickness, lateral and axial resolution, voxel size, number of frames per stack etc.

4) Besides showing optical XY sections and maximum intensity projections, the authors should add orthogonal XZ or YZ views in figures 2 and 4 (and/or in supplemental figures) as necessary to show the actin/membrane/nuclear signal at different depths of cells and tissues during gastrulation of the honeybee embryo.

5) In the section "Honeybee EEM formation in an evolutionary context", the authors report a larger variability in the closure site of the extraembryonic membrane of the honeybee embryo compared to the beetle embryo (SD 7.3% vs. 3.1%). Based on this finding, they argue that the physical interaction between the embryo and the vitelline membrane (previously identified in beetle and fly embryos) does not operate in honeybees. I suggest that the authors remove this premature conclusion as there are currently no data in favour or against this mechanism in honeybees (that also exhibit extensive asymmetric tissue movements). Furthermore, the connection between the precision in EEM closure and subsequent degradation is entirely speculative and could be also omitted.

6) For clarity, show embryo length in the x-axis of Fig. 3D in percentages rather than actual μ m along the AP axis.

7) The authors should cite at the end of the introduction the papers by Benton et al. 2013 (PMID: 23861059), Benton 2018 (PMID: 29969459) and Jain et al. 2020 (PMID: 33154375) regarding the use of fluorescence live imaging and image analysis to study EEM dynamics in *Tribolium*, as well as the paper by Hörl et al. 2019 (PMID: 31384047) in the M&M section "Image data processing" in case they used the Fiji/Imagej plugin BigStitcher for tile stitching.

8) In the "Perspectives" section, the authors should discuss the limitations - if any - of their labelling and imaging methodologies and potential improvements that could be implemented in the future.

Author response to reviewers' comments

Reviewer #1

Concern #1-1

I initially reviewed this manuscript and came to the two conclusions further below, but before submitting my recommendation, I decided to download some of the authors' data from zenodo to see if I could recommend publishing it as a dataset. The very first dataset I looked at (DS0004 DOI 10.5281/zenodo.13946275) showed that the authors claim that their method "does not impair development" is simply not true. This dataset is the only dorsal view shared by the authors and it shows essentially all of the dorsal and anterior tissue disintegrating. This phenotype is so obvious, it is concerning that the authors chose to ignore it. Furthermore, this is the only dorsal view on zenodo, and also the only view not provided as a supplementary

movie with the manuscript. I attach a movie of the timelapse with this review. I did not invest the time to check every timelapse on zenodo, but a closer look at Supp Vid 2 also shows defects at the anterior pole of the egg.

Tissue defects during live imaging happen, especially when establishing a new system, but trust is essential in academia. One must be able to trust that when scientists make statements, and present data to support those statements, they are being truthful. Unfortunately, that is clearly not the case here.

Answer #1-1

We agree with the reviewer's observation that the embryo in dataset DS0004 (DOI 10.5281/zenodo.13946275) exhibits developmental aberrations and are grateful for this valuable feedback. However, we respectfully object to the assertion that we "chose to ignore" this phenotype. This was not a deliberate omission but an unintended oversight in preparing the manuscript and figures. We would like to clarify with four points:

- (i) Our data management is highly automated; for example, scripts systematically extract single maximum projections from time stacks for figure assembly. DS0004 is the least-used dataset in our study and was not included in any quantitative analyses (e.g., Figure 3). Only one image from this dataset appears in Figure 2A, serving merely as an example of a dorsal view. We regret not noticing the tissue defects during figure preparation and acknowledge that we should have inspected the dataset more carefully. The single image from Figure 2A has been removed, and DS0004 is now only mentioned in the "Limitations of our framework" section in the context of live imaging survival rates.
- (ii) Prior to peer review and without request, we deposited all datasets on Zenodo to make them openly accessible to reviewers. This reflects our commitment to transparency and open science. The datasets are organized for direct inspection without further processing, which we believe facilitates constructive peer review.
- (iii) Regarding Supplementary Video 2, we respectfully disagree that the embryo shows defects at the anterior pole. Upon re-examination, we confirm that the embryo depicted hatches morphologically intact. We would be happy to provide additional frames or analyses for clarification.
- (iv) We agree that trust and rigorous data evaluation are essential in science. We also believe that peer review is precisely the mechanism by which oversights like ours can be identified and corrected before publication, and this case could serve as a prime example of constructive peer view. However, the reviewer's conclusion that we were not being truthful is unwarranted and not supported by the facts. There was no intention to misrepresent our results.

Concern #1-2

First, while this article is submitted as a "Research Report", it is not clear what, if any, research question is answered. The authors state that live imaging is superior to previous fixed light and electron microscopy methods for examining development and morphogenesis, yet they do not demonstrate that benefit in this work. Rather, work from 50 to over 100 years ago (Nelson, 1915; Fleig and Sander 1986; Fleig and Sander, 1988; Du Praw 1967) give far more comprehensive and informative reports of Apis embryogenesis, entirely using fixed methods. Netschitailo et al themselves point this out.

Answer #1-2

We thank the reviewer for this important comment. We would like to clarify that we originally intended to submit this manuscript to the "Techniques & Resources" section, but mistakenly

selected a different category during online submission; we apologize for the confusion.

Our study does not seek to replicate classical descriptions of honeybee embryogenesis, but rather to establish the first 4D fluorescence live imaging protocol. This approach enables continuous, high-resolution visualization of dynamic developmental processes, providing insights that fixed-stage methods cannot (as illustrated in Figure 3).

Whereas earlier studies offered comprehensive static descriptions, live imaging captures temporal continuity and avoids interpretative gaps between fixed time points. Our work lays the technical foundation for such analyses in the honeybee and delivers a systematic, time-resolved dataset that is openly available to the community for further exploration and hypothesis-driven research.

We have also strengthened the introduction to better emphasize the value of fluorescence live imaging approaches.

Concern #1-3

Second, if one considers the manuscript for technical advancements, it also falls short. Most of the results section consists of the authors' descriptions of their use of previously established techniques, yet generally without citing previous work:

- embryonic microinjection as reported in Apis for many years - not cited here

- transient expression for live imaging as reported in many species, e.g. fish, cnidarians, spiders, and beetles, - not cited here

- halocarbon oil for imaging as used for many species, with this particular example also used for plants,

- a lightsheet microscope that the authors mention no details of, but is presumably the same they have used in multiple previous reports.

The genuinely new advances are:

- a custom mounting chamber described in some detail, although this is the norm for the field of light sheet microscopy

- new plasmids for transient expression and live imaging, although the authors do not describe how these tools improve on those previously created for use in other insects

Answer #1-3

We would like to respond to the list of arguments as follows:

- (i) We added citations on embryonic microinjection in honeybees wherever appropriate and expanded the references already provided in the Methods.
- (ii) We agree that a brief overview of mRNA injection-based labeling in other model organisms strengthens the manuscript and have added a corresponding section to the Introduction.
- (iii) We acknowledge that usage of PFD was not properly justified and have now included a brief explanation, particularly highlighting how it differs from “classical” halocarbon oils.
- (iv) While the light sheet fluorescence microscope setup was illustrated in the metadata tables, we agree this information appeared somewhat hidden. We therefore added a short dedicated description in the Methods section.
- (v) We have more clearly embedded the chamber design into the broader context of the study.
- (vi) We now emphasize the *raison d'être* of our plasmids in greater detail throughout the

manuscript, with a focus on shareability and optimization.

Reviewer #2

Concern #2-1

The manuscript is very well written and contains most of the details I would like to see in such a publication. There are a couple of things I would be keen to see added to help practitioners in this space take up this technology. 1) A table of survival rates of their various treatments, in the supplemental materials. While there are details of survival rate for the various solutions used, it would be useful to bring all of that data together. The expected survival rates are hugely useful for anyone repeating these experiments.

Answer #2-1

We have compiled the survival rates in a new supplementary table (now: Supplementary Table 3) and ensured that all numbers (ratios) are traceable throughout the manuscript, making their origin fully transparent. In the extended Discussion, we now also emphasize that survival rates must be considered in combination and sequence across the entire experimental procedure—from injection through incubation and mounting to imaging in the light sheet microscope.

Concern #2-2

2) A diagram of the humid chambers used to incubate injected embryos would be useful- there is quite a lot of disagreement as to what is the best solution for honeybee embryos. The authors obviously have found a solution, and it would be good to share that.

Answer #2-2

We were not aware that this would be a detail of interest, as the method is already used in many publications (including those mentioned in the Methods section). We are pleased to be able to provide more information here and added an image to the respective supplementary figure (now: Supplementary Figure 8, image D). The methods section additionally lists the air volume of the boxes used and the dosage of sulfuric acid to prevent moulding and ensure a sufficiently humidified atmosphere.

Reviewer #3

Concern #3-1

My major comment is that this study is best suited for the Techniques and Resources section rather than the Research Articles/Reports section of Development. Most of the paper describes the technical aspects (embryo microinjection protocol, mounting medium and customization of the imaging chamber) that had to be optimized for live imaging of honeybee embryogenesis. The cell and tissue dynamics accompanying honeybee embryogenesis are not extracted from these image datasets, except for a brief comparison of the precision in the closure sites of the extraembryonic membranes between honeybee and beetle embryos (see also point 5 below).

Answer #3-1

As explained in Answer #1-2, we initially intended to submit our study as contribution to the *Development* “Techniques & Resources” section. On further notice, this concern is basically answered by the editorial assignment to the *Biology Open* “Techniques & Resources” section.

Concern #3-2

1) *In the section "mRNA-injection-based fluorescence labelling of honeybee embryos", please indicate the time of injection and the time of blastoderm cellularization (in hours after egg lay) as these are crucial parameter for homogeneous embryo labelling in transient labelling protocols.*

Answer #3-2

We included the time of injection (00:30-03:00 h after egg laying) and the blastoderm cellularization timing as described in literature to the "mRNA-injection based fluorescence labeling of honeybee embryos" section of the Results.

Concern #3-3

2) *In the same section or in the M&M section "Injection of honeybee embryos" explain why you chose a single mRNA concentration of 600 ng/μl for injections and whether you tried other concentrations for stronger and/or more homogeneous labelling. In case of multilabel experiments such as the one shown in Fig. 4, please specify in M&M whether each mRNA was injected at 600 ng/μl or at different concentrations. Please give also an estimate of the injected volume per embryo with the needles and microinjection set up used.*

Answer #3-3

The used mRNA concentration of 600 ng/μl in the injected volume of ~ 400 pl results in a total amount of ~240 pg mRNA per embryo. In the work published by Otte et al. (2018) 240 pg of transposase-encoding mRNA was safely used for genetic manipulation studies in embryos of the same age (Injection 1:30 h AEL) with a similar setup. Therefore, we could assume that this concentration resulted in viable embryos and used it for our tests. Since we had sufficient signal in the recordings, we did not test any higher concentrations. Each mRNA was adjusted to a concentration of 600 ng/μl before injection. The details on this were added to the method section.

Concern #3-4

3) *In the section "Long-term fluorescence live imaging of honeybee embryos", specify that live imaging was performed with single-sided illumination, single-sided detection and with a temporal resolution of 30 min. This information is currently scattered in different parts of the manuscript. Importantly, in the M&M section "Honeybee and beetle embryo mounting and microscopy", give all necessary details about the imaging parameters, such as illumination and detection objectives used, light-sheet thickness, lateral and axial resolution, voxel size, number of frames per stack etc.*

Answer #3-4

This concern relates in part to Concern #1-3, which also criticized that the information regarding the microscope setup is hidden. We expanded both the Materials and Methods section considerably by adding a moderately sized description of the microscope setup and central imaging parameters, and updated the Supplementary Metadata table to contain all requested information.

Concern #3-5

4) Besides showing optical XY sections and maximum intensity projections, the authors should add orthogonal XZ or YZ views in figures 2 and 4 (and/or in supplemental figures) as necessary to show the actin/membrane/nuclear signal at different depths of cells and tissues during gastrulation of the honeybee embryo.

Answer #3-5

We added two more supplementary figures (now: Supplementary Figures 5 (with relates to Figure 2) and 7 (which relates to Figure 4)) that contain orthogonal views.

Concern #3-6

5) In the section "Honeybee EEM formation in an evolutionary context", the authors report a larger variability in the closure site of the extraembryonic membrane of the honeybee embryo compared to the beetle embryo (SD 7.3% vs. 3.1%). Based on this finding, they argue that the physical interaction between the embryo and the vitelline membrane (previously identified in beetle and fly embryos) does not operate in honeybees. I suggest that the authors remove this premature conclusion as there are currently no data in favour or against this mechanism in honeybees (that also exhibit extensive asymmetric tissue movements). Furthermore, the connection between the precision in EEM closure and subsequent degradation is entirely speculative and could be also omitted.

Answer #3-6

We agree with the reviewer that there is currently too little evidence for this statement as written. Since we initially submitted our manuscript to *Development* as in the "Research report" format, in which the Results and Discussion sections are combined, the statement had a strong conclusive character. Since now, the combined section is split into two parts - one for the Results section and one for the Discussion section - for re-submission to Biology Open, we decided to keep the statement, considerably weakened, in the form of a theory in the Discussion section.

Concern #3-7

6) For clarity, show embryo length in the x-axis of Fig. 3D in percentages rather than actual μm along the AP axis.

Answer #3-7

We adapted the figure accordingly.

Concern #3-8

7) The authors should cite at the end of the introduction the papers by Benton et al. 2013 (PMID: 23861059), Benton 2018 (PMID: 29969459) and Jain et al. 2020 (PMID: 33154375) regarding the use of fluorescence live imaging and image analysis to study EEM dynamics in *Tribolium*, as well as the paper by Hörl et al. 2019 (PMID: 31384047) in the M&M section "Image data processing" in case they used the Fiji/ImageJ plugin BigStitcher for tile stitching.

Answer #3-8

We appreciate the suggestion from Reviewer #3 and added a citation regarding the suggested studies to the end of the introduction's second paragraph. Stitching was actually performed with the Fiji plugin by Preibisch et al. 2009, and is properly cited in the Image data processing sub-section.

Concern #3-9

8) In the "Perspectives" section, the authors should discuss the limitations - if any - of their labelling and imaging methodologies and potential improvements that could be implemented in the future.

Answer #3-9

Within the Discussion, we do now address the following limitations (and provide improvement ideas if applicable):

- (i) improvement potential regarding survival
 - (ii) mRNA-based labeling allows ubiquitous labeling only
 - (iii) assessment limited to female embryos
 - (iv) throughput
-

Second decision letter

MS ID#: bio.062151

MS Title: Long-term fluorescence live imaging of honeybee embryos using light sheet fluorescence microscopy and halocarbon-based liquids

Authors: Oksana Netschitailo, Paul Siefert, Markus M. Middeke, Artemiy Golden, Christin Schrod, Tim Beringer, Bernd Grünewald, Ernst H.K. Stelzer and Frederic Strobl

I am happy to tell you that your manuscript has been accepted for publication in Biology Open, pending our standard publication integrity checks. It was accepted on 26th August 2025.